# Arginase 1 is a key driver of immune suppression in pancreatic cancer

Rosa E Menjivar[1], Zeribe C Nwosu[2], Wenting Du[3], Katelyn L Donahue[4], Hanna S Hong[5], Carlos Espinoza[3], Kristee Brown[3], Ashley Velez-Delgado[6], Wei Yan[3], Fatima Lima[3], Allison Bischoff[4], Padma Kadiyala[5], Daniel Salas-Escabillas[4], Howard C Crawford[7], Filip Bednar[3,8], Eileen Carpenter[8,9], Yaqing Zhang[3,8], Christopher J Halbrook[10,11], Costas A Lyssiotis[2,4,8,9]*, Marina Pasca di Magliano[1,4,6,7,8]*

[1]Cellular and Molecular Biology Program, University of Michigan-Ann Arbor, Ann Arbor, United States; [2]Department of Molecular and Integrative Physiology, University of Michigan-Ann Arbor, Ann Arbor, United States; [3]Department of Surgery, University of Michigan-Ann Arbor, Ann Arbor, United States; [4]Cancer Biology Program, University of Michigan-Ann Arbor, Ann Arbor, United States; [5]Department of Immunology, University of Michigan-Ann Arbor, Ann Arbor, United States; [6]Department of Cell and Developmental Biology, University of Michigan-Ann Arbor, Ann Arbor, United States; [7]Henry Ford Pancreatic Cancer Center, Detroit, United States; [8]Rogel Cancer Center, Ann Arbor, United States; [9]Department of Internal Medicine, Division of Gastroenterolog, University of Michigan-Ann Arbor, Ann Arbor, United States; [10]Department of Molecular Biology and Biochemistry, University of California, Irvine, Irvine, United States; [11]Chao Family Comprehensive Cancer Center, University of California, Irvine, Irvine, United States

*For correspondence:
clyssiot@umich.edu (CAL);
marinapa@umich.edu (MPdM)

**Abstract** An extensive fibroinflammatory stroma rich in macrophages is a hallmark of pancreatic cancer. In this disease, it is well appreciated that macrophages are immunosuppressive and contribute to the poor response to immunotherapy; however, the mechanisms of immune suppression are complex and not fully understood. Immunosuppressive macrophages are classically defined by the expression of the enzyme Arginase 1 (ARG1), which we demonstrated is potently expressed in pancreatic tumor-associated macrophages from both human patients and mouse models. While routinely used as a polarization marker, ARG1 also catabolizes arginine, an amino acid required for T cell activation and proliferation. To investigate this metabolic function, we used a genetic and a pharmacologic approach to target *Arg1* in pancreatic cancer. Genetic inactivation of *Arg1* in macrophages, using a dual recombinase genetically engineered mouse model of pancreatic cancer, delayed formation of invasive disease, while increasing CD8+ T cell infiltration. Additionally, *Arg1* deletion induced compensatory mechanisms, including *Arg1* overexpression in epithelial cells, namely Tuft cells, and *Arg2* overexpression in a subset of macrophages. To overcome these compensatory mechanisms, we used a pharmacological approach to inhibit arginase. Treatment of established tumors with the arginase inhibitor CB-1158 exhibited further increased CD8+ T cell infiltration, beyond that seen with the macrophage-specific knockout, and sensitized the tumors to anti-PD1 immune checkpoint blockade. Our data demonstrate that Arg1 drives immune suppression in pancreatic cancer by depleting arginine and inhibiting T cell activation.

## Editor's evaluation

Menjivar et al. identify an important, previously unrecognized role of myeloid cell Arginase1 (Arg1) activity in shaping the anti-tumor immune response in pancreatic ductal adenocarcinoma (PDAC). The proposed therapeutic combination is a convincing new approach for pancreatic cancer, with an enhanced response to immune therapy upon arginase inhibition.

## Introduction

Pancreatic ductal adenocarcinoma (PDA) is currently the third leading cause of cancer-related deaths in the United States with a 5-y survival rate of 12% (*Siegel et al., 2023*). This poor survival rate is due to late detection and ineffective treatments. The hallmark mutation in PDA is found in the *KRAS* gene, most commonly *KRAS*$^{G12D}$ (*Hezel et al., 2006*; *Hingorani et al., 2003*; *Hingorani et al., 2005*; *Schneider and Schmid, 2003*), whereas disease progression is accelerated by loss of tumor suppressor genes, including TP53, SMAD4, and INK4A (*Hezel et al., 2006*; *Maitra and Hruban, 2008*).

Standard of care for PDA is a combination of systemic chemotherapy that includes FOLFIRINOX or gemcitabine plus nab-paclitaxel (*Mizrahi et al., 2020*), which provides limited improvement in patient survival. Oncogenic KRAS proteins were long thought to be 'undruggable' (*Gysin et al., 2011*; *Liu et al., 2019*). Recently, new approaches to drug discovery have led to the development of KRAS$^{G12C}$ inhibitors (*Canon et al., 2019*; *Fell et al., 2020*), a common mutation in non-small cell lung cancer (*Prior et al., 2012*); initial clinical data show not only dramatic responses but also cases of primary or secondary resistance (*Awad et al., 2021*). While the applicability of G12C inhibitors is limited in PDA, which is rarely present with this specific allele (*Witkiewicz et al., 2015*), new KRASG12D inhibitors have recently been described and are entering the clinic (*Hallin et al., 2022*; *Wang et al., 2022*). Based on the observations in lung cancer (*Awad et al., 2021*; *Zhao et al., 2021*) and on predictions based on mouse models that allow genetic inactivation of oncogenic KRAS (*Collins et al., 2012*; *Ying et al., 2012*), it is to be expected that resistance to KRAS G12D inhibitors will similarly arise; activation of anti-tumor immune responses remains our best hope for long-term tumor control. Although immune checkpoint inhibitors are effective in other cancers, this benefit has not translated to PDA (*Brahmer et al., 2012*; *Royal et al., 2010*) due to the severely immunosuppressive tumor microenvironment (TME) that characterizes this disease (*Morrison et al., 2018*; *Vonderheide and Bayne, 2013*). Approaches combining immune checkpoints and myeloid cell targeting might be more promising, based on preclinical studies (*DeNardo et al., 2021*; *Gulhati et al., 2023*; *Liu et al., 2021*). Here, we explore new approaches targeting macrophages to induce anti-tumor immunity in pancreatic cancer.

The pancreatic TME includes cancer-associated fibroblasts and a heterogenous population of immune cells, the majority of which is myeloid cells (*Gabrilovich and Nagaraj, 2009*). These myeloid cells include tumor-associated macrophages (TAMs), immature myeloid cells, also referred to as myeloid-derived suppressor cells, and granulocytes, such as neutrophils (*Gabrilovich et al., 2012*). In contrast, CD8$^+$ T cells are rare in the pancreatic TME, although their prevalence is heterogeneous in different patients (*Stromnes et al., 2017*). Single-cell RNA sequencing (sc-RNA-seq) analysis revealed that most CD8$^+$ T cells infiltrating PDA have an exhausted phenotype (*Steele et al., 2020*). An understanding of the mechanisms mediating immune suppression in pancreatic cancer is needed to design new therapeutic approaches for this disease. Of note, analysis of long-term survivors revealed persistence of tumor-specific memory CD8$^+$ T cells, indicating that when an anti-tumor immune response does occur, it leads to effective tumor control (*Balachandran et al., 2017*). Conversely, evidence of loss of antigens over time due to immunoediting has also been described (*Łuksza et al., 2022*). Furthermore, myeloid cell depletion in mouse models of pancreatic cancer led to activation of anti-tumor T cell responses (*Mitchem et al., 2013*; *Zhang et al., 2017a*), spurring an effort to target myeloid cells in pancreatic cancer (*Nywening et al., 2016*); yet, clinical efficacy has been low, and more effective approaches are needed.

Myeloid cells infiltrating the neoplastic pancreas, and ultimately those in PDA, express high levels of Arginase 1 (*Arg1*). ARG1 is one of the enzymes that metabolize arginine, and it is prevalently located in the cytoplasm, while the ARG2 isoform is prevalently mitochondrial (*Bronte and Zanovello, 2005*). Indeed, we previously showed that *Arg1* expression in myeloid cells is driven by oncogenic *Kras* expression/signaling in epithelial cells, starting during early stages of carcinogenesis (*Velez-Delgado et al., 2022*; *Zhang et al., 2017b*). Beyond our work, ARG1 is widely appreciated as a

marker of alternatively polarized macrophages. An increase in tumor myeloid *Arg1* expression has been reported in other cancers, including renal cell carcinoma (*Rodriguez et al., 2009*), breast (*Polat et al., 2003*; *Singh et al., 2000*), colon (*Arlauckas et al., 2018*), and lung cancer (*Miret et al., 2019*).

In addition to its role as a marker of alternatively polarized macrophages, ARG1 is a metabolic enzyme that breaks down the amino acid L-arginine to urea and ornithine (*Jenkinson et al., 1996*). Connecting this activity to myeloid Arg1 expression, several reports detail how arginine is necessary for the activation and proliferation of CD8 T cells (*Rodriguez et al., 2007*; *Rodriguez et al., 2004*). This indicates that myeloid cells may deplete arginine in the TME to dampen anti-tumor T cell activity. Based on these concepts, a small-molecule arginase inhibitor, CB-1158 (INCB001158) (Calithera Biosciences, Inc, South San Francisco, CA), was developed. CB-1158 treatment as monotherapy, or in combination with anti-PD-1 checkpoint inhibitor, decreased tumor growth in vivo in mice with Lewis lung carcinoma (*Steggerda et al., 2017*). CB-1158 also inhibits human arginase, and it is being tested in a phase I clinical trial in patients with advanced or metastatic solid tumors (*Pham et al., 2018*; *Steggerda et al., 2017*).

Encouraged by these studies and the prominent disease-specific expression of Arg1 in PDA macrophages, we set forth to determine its functional role. Here, we used a dual-recombinase approach to delete *Arg1* in myeloid cell lineages via Cre-loxP technology and at the same time induced oncogenic *Kras* in pancreatic epithelial cells using the orthogonal Flp-Frt recombination approach. We discovered that *Arg1* deletion in myeloid cells profoundly reshaped the TME, increased the infiltration of CD8+ T cells, and reduced malignant disease progression. However, we also noticed a resistance mechanism whereby other cells in the TME, such as epithelial cells, upregulate *Arg1* expression, potentially blunting the effect of its inactivation in myeloid cells. We thus employed a systemic approach, whereby we treated a syngeneic orthotopic model of pancreatic cancer with the arginase inhibitor CB-1158 and found it to sensitize PDA to anti-PD1 immune checkpoint blockade. These results illustrate a functional role of Arg1 in pancreatic tumor-derived myeloid cells, reveal novel aspects of intratumoral compensatory metabolism, and provide new inroads to increase the efficacy of checkpoint immune therapy for PDA.

## Results

### Pancreatic cancer infiltrating myeloid cells express Arginase 1

We have previously reported that expression of oncogenic *Kras* in pancreas epithelial cells drives expression of *Arg1* in macrophages in vivo during early stages of carcinogenesis (*Velez-Delgado et al., 2022*; *Zhang et al., 2017b*). We sought to determine whether Arg1 was also associated with late stages of carcinogenesis in mouse and human tumors. As there was no validated antibody for human ARG1 available, we performed *ARG1* RNA in situ hybridization in human PDA, together with co-immunofluorescence staining for the immune cell marker CD45 and for the epithelial cell marker E-cadherin (ECAD) (*Figure 1A*). We observed prevalent *ARG1* expression in CD45+ cells, and occasional low expression in ECAD+ cells (*Figure 1A*, *Figure 1B*). Analysis of a previously published sc-RNA-seq dataset, which includes 16 human PDA samples (*Steele et al., 2020*; *Figure 1C* and *Figure 1—figure supplement 1A*), revealed highest expression of *ARG1* in myeloid cells (*Figure 1D*, *Figure 1—figure supplement 1B*). In contrast, we observed minimal *ARG1* expression in CD4+ T cells, CD8+ T cells, epithelial cells, and fibroblasts (*Figure 1D*). *ARG2* was expressed at high levels in endocrine cells, and at lower, but detectable levels in epithelial cells (*Figure 1—figure supplement 1B*). Thus, we conclude that myeloid cells are the main source of *ARG1* in the pancreatic cancer microenvironment.

Infiltration of myeloid cells, specifically macrophages, in pancreatic cancer portends worse patient survival (*Sanford et al., 2013*; *Tsujikawa et al., 2017*). We thus assessed whether *ARG1* expression correlated with worse patient outcomes. Based on a publicly available human PDA microarray data set (GSE71729) (*Moffitt et al., 2015*), we found that high *ARG1* expression in pancreatic cancer correlated with worse survival (*Figure 1E*), suggesting that *ARG1* in myeloid cells may play a functional role in human PDA.

Next, to determine whether the increase in *Arg1* expression was recapitulated in a mouse model of pancreatic cancer, we analyzed our mouse sc-RNA-seq data from healthy mice and tumor-bearing KPC (*Kras*[LSL-G12D/+];*p53*[LSL-R172H/+];*Ptf1a*[Cre/+]) (*Hingorani et al., 2005*) mice (*Figure 1F* and *Figure 1—figure supplement 1C and D*). We detected the highest level of *Arg1* expression in macrophages

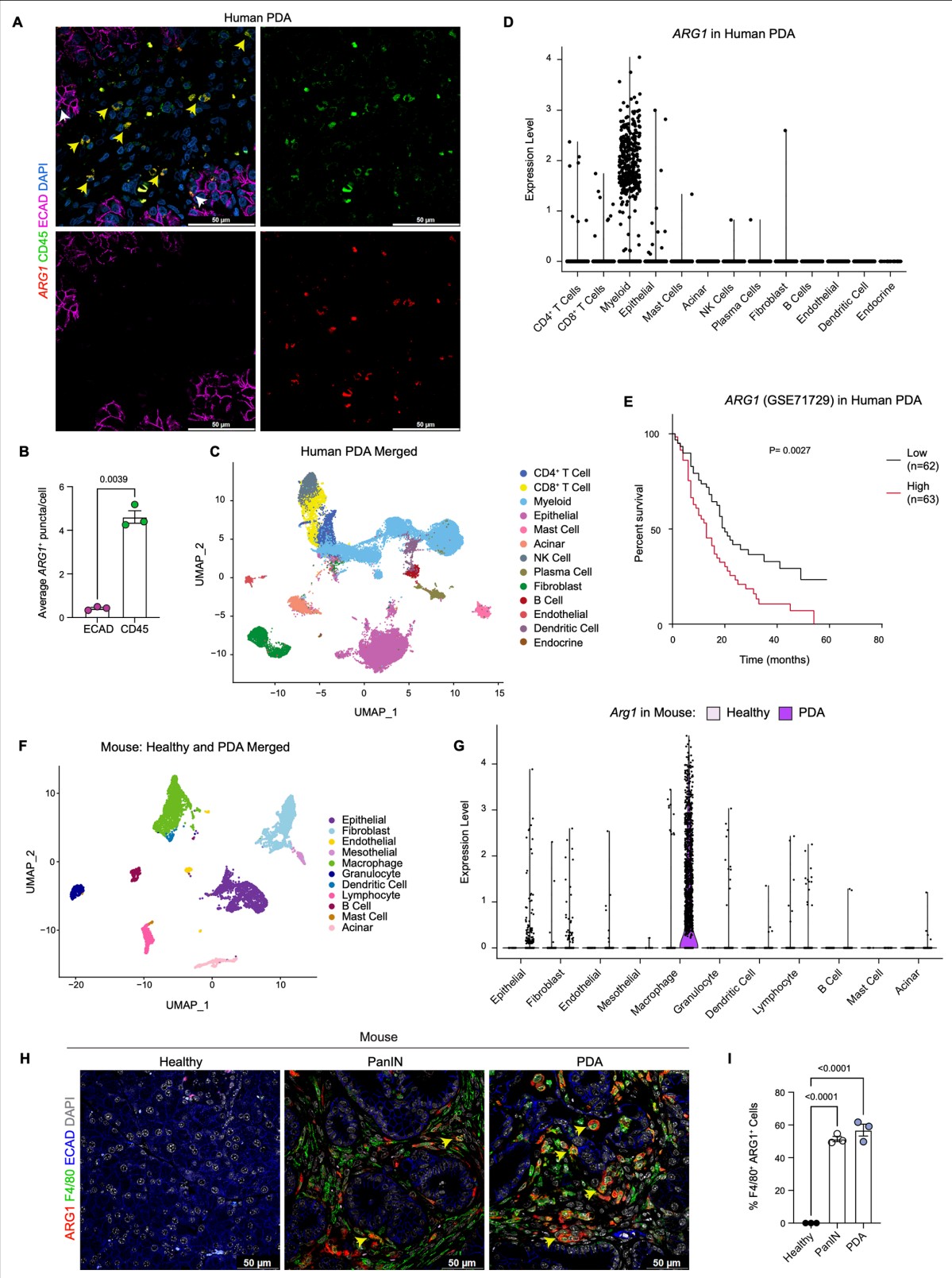

**Figure 1.** Arginase 1 (Arg1) is highly expressed in human and mouse myeloid cells. (**A**) Representative image of RNA in situ hybridization (ISH) of *ARG1* (red) and co-immunofluorescence staining of immune (CD45, green) and epithelial (E-cadherin [ECAD], magenta) cells in human pancreatic ductal adenocarcinoma (PDA). Counterstain, DAPI (blue). White arrows point to *ARG1*-ISH in ECAD+ cells, and yellow arrows point to *ARG1*-ISH in CD45+ cells. Scale bar, 50 µm. (**B**) Quantification of average *ARG1*+ puncta in CK19 and CD45 cells. Unpaired t test with Welch's correction was used to determine

*Figure 1 continued on next page*

Figure 1 continued

statistical significance. (C) Uniform manifold approximation and projection (UMAP) visualization of 13 identified cell populations from single-cell RNA sequencing (sc-RNA-seq) of 16 human PDA tumors. Data from *Steele et al., 2020*. (D) Violin plot of normalized gene expression of *ARG1* in the identified cell populations from the human PDA sc-RNA-seq. (E) Survival analysis of a human PDA microarray data set (GSE71729) with low (n=62) and high (n=63) *ARG1* expression. Statistical significance was determined using the Kaplan Meier overall survival Logrank test. (F) UMAP visualization of 11 identified populations from healthy and PDA merged mouse sc-RNA-seq. (G) Violin plot of normalized gene expression of *Arg1* in the identified cell populations from mouse sc-RNA-seq. (H) Representative co-immunofluorescence staining for ARG1 (red), macrophages (F4/80, green), and epithelial (ECAD, blue) cells in mouse tissue at different stages of disease. Counterstain, DAPI (gray). Scale bar, 50 μm. Yellow arrows indicate ARG1 expression in F4/80$^+$ cells. (I) Quantification of F4/80$^+$ ARG1$^+$ cells in healthy, PanIN, and PDA mouse tissue. Statistical significance was determined using an ordinary one-way ANOVA with multiple comparisons. p Value was considered statistically significant when p<0.05.

The online version of this article includes the following figure supplement(s) for figure 1:

**Figure supplement 1.** Arginase 1 (Arg1) expression in human and mouse myeloid cells.

(*Figure 1G* and *Figure 1—figure supplement 1E*). Importantly, *Arg1* expression was enriched in PDA infiltrating macrophages compared to macrophages in the normal pancreas (*Figure 1G* and *Figure 1—figure supplement 1E*). Other cell types, such as epithelial cells and fibroblasts, only had sporadic *Arg1* expression. *Arg2* was also expressed in macrophages, albeit at low levels, and was the prevalent isoform in tumor-associated granulocytes (*Figure 1—figure supplement 1E*). To evaluate the protein expression of ARG1, we performed co-immunofluorescence staining. We included healthy pancreas samples, PanIN-bearing pancreata from KC mice (*Kras$^{LSL-G12D/+}$;Ptf1a$^{Cre/+}$*) (*Hingorani et al., 2003*), and PDA. We did not detect ARG1 protein in the healthy pancreas. In contrast, both PanIN and PDA presented with frequent co-localization of the macrophage marker F4/80 with ARG1, consistent with prevalent expression in this cell population (*Figure 1H and I* and *Figure 1—figure supplement 1F*). Consistent with human data, expression of ARG1 in other cell types was rare (*Figure 1H* and *Figure 1—figure supplement 1F*). Thus, both in mouse and human pancreatic cancer, ARG1 is highly expressed and largely confined to tumors, where it is predominantly expressed in macrophages.

## Arginase 1 deletion in myeloid cells reduces tumor progression and induces macrophage repolarization

To examine the function of myeloid *Arg1* in PDA, we generated mice lacking *Arg1* expression in myeloid cells. Specifically, we crossed *Arg1$^{f/f}$* mice with *Lyz2$^{Cre/+}$* mice to generate *Lyz2$^{Cre/+}$;Arg1$^{f/f}$* mice (*El Kasmi et al., 2008*). *Lyz2$^{Cre/+}$* mice were generated by inserting the Cre cDNA in the endogenous M lysozyme (LysM) locus (*Clausen et al., 1999*), which is broadly expressed in myeloid cells, including macrophages and neutrophils (*Clausen et al., 1999; El Kasmi et al., 2008*). To validate the deletion of *Arg1* in macrophages, we harvested bone marrow (BM) cells from wild type (WT) or *Lyz2$^{Cre/+}$;Arg1$^{f/f}$* mice and cultured these directly in pancreatic cancer cell conditioned media (CM) for 6 d. This protocol yields BM-derived TAMs (*Figure 2A*), as previously described (*Zhang et al., 2017a*). ARG1 expression was readily detectable in WT TAMs; in contrast, *Lyz2$^{Cre/+}$;Arg1$^{f/f}$* TAMs had none or very low ARG1 protein, indicating efficient Cre recombination (*Figure 2B*). We then sought to determine whether ARG1 expression in TAMs affected the metabolite composition of the media. We thus performed extracellular metabolomics by liquid chromatography couple tandem mass spectrometry (LC-MS/MS) and observed elevated levels of L-Arginine in the *Lyz2$^{Cre/+}$;Arg1$^{f/f}$* TAM medium compared with WT TAM medium (*Figure 2C*). This finding is consistent with WT TAMs depleting arginine from their growth medium at an enhanced rate, relative to TAMs lacking *Arg1* expression.

To investigate the function of *Arg1* in myeloid cells during PDA progression, we generated *Kras$^{Frt-STOP-Frt-G12D/+}$;Ptf1a$^{FlpO/+}$* (KF); *Lyz2$^{Cre/+}$;Arg1$^{f/f}$* mice, hereafter referred to as KFCA (*Figure 2D* and *Figure 2—figure supplement 1A*). The dual-recombinase system integrates both Flippase-FRT (Flp-FRT) and Cre-loxP recombination technologies to independently modify epithelial cells and myeloid cells (*Garcia et al., 2020; Wen et al., 2019*). We aged a cohort of KF and KFCA mice for 2 mo, a timepoint at which PanIN lesions are formed in KF mice (*Figure 2—figure supplement 1A*). To evaluate the efficiency of *Arg1* deletion, we performed co-immunofluorescent staining for ARG1, F4/80, and CK19. As expected, we observed abundant expression of ARG1 in F4/80$^+$ macrophages in KF, and minimal to no expression of ARG1 in macrophages from KFCA tissue (*Figure 2—figure supplement 1B*). We then performed histopathological evaluation of the tissue to determine the functional effect of loss of myeloid *Arg1*. Here, we observed a trending decrease in ADM and PanIN lesions (although

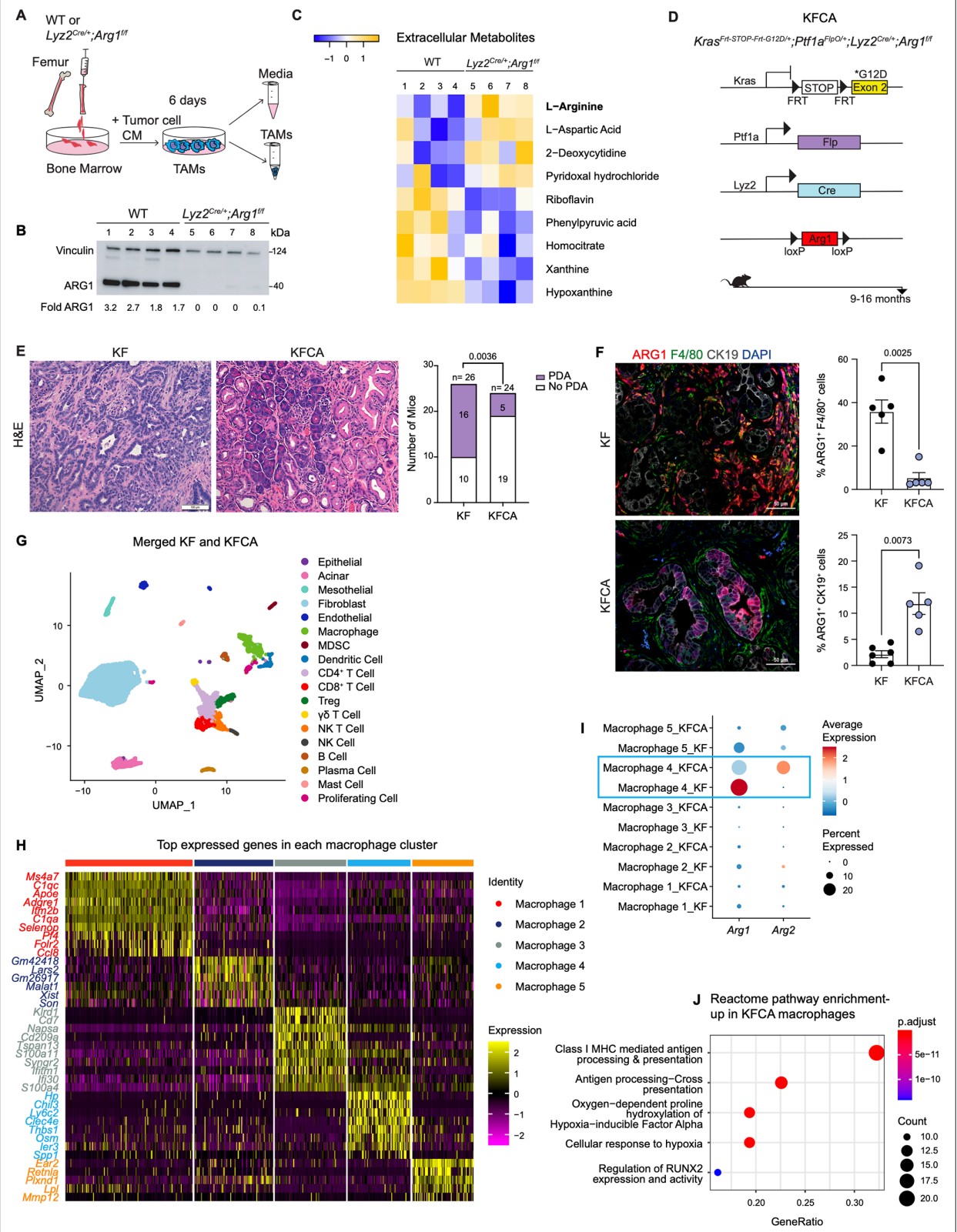

**Figure 2.** Arginase 1 (*Arg1*) deletion in myeloid cells reduces tumor progression and induces macrophage repolarization in a spontaneous pancreatic ductal adenocarcinoma (PDA) mouse model. (**A**) Schematic illustration for the generation of tumor-associated macrophages (TAMs) from wild-type (WT) and *Lyz2^Cre/+^;Arg1^f/f^* mice. (**B**) Representative image of western blot for ARG1 levels in WT and *Lyz2^Cre/+^;Arg1^f/f^* TAMs. Vinculin, loading control. (**C**) Heatmap of statistically significantly different extracellular metabolites from WT (lanes 1–4) and *Lyz2^Cre/+^;Arg1^f/f^* (lanes 5–8) TAM media. (**D**) Genetic

*Figure 2 continued on next page*

*Figure 2 continued*

makeup of the $Kras^{Frt-STOP-Frt-G12D/+}$;$Ptf1a^{FlpO/+}$;$Lyz2^{Cre/+}$;$Arg1^{f/f}$ (KFCA) mouse model for the deletion of *Arg1* in myeloid cells during PDA tumorigenesis. Data shown here from mice aged 9–16 mo, n=24–26/group. (**E**) Representative hematoxylin and eosin (H&E) staining from age matching KF and KFCA mice. Scale bar, 100 μm. Histopathology evaluation shown on the right (n=26 KF and 24 KFCA, age matched, 9–16 mo old). Statistical significance was determined using chi-square test. Statistically significant when *P*<0.05. (**F**) Representative image of co-immunofluorescence staining for ARG1 (red), macrophages (F4/80, green), epithelial (CK19, gray), and DAPI (blue) in KF and KFCA tissue. Scale bar, 50 μm. Quantification on the right, n=5–6/group. Significance was determined using unpaired t test with Welch's correction. Statistically significant when p<0.05. (**G**) Uniform manifold approximation and projection (UMAP) visualization for the identified cell populations in merged KF and KFCA single-cell RNA sequencing. (**H**) Heatmap of top differentially expressed genes in the macrophage subclusters identified from KF and KFCA pancreata. (**I**) Dot plot visualization of *Arg1* and *Arg2* expression in KF and KFCA macrophage clusters. Average expression is shown by color intensity and expression frequency by dot size. (**J**) Reactome pathway enrichment analysis showing significantly upregulated pathways in KFCA macrophages.

The online version of this article includes the following source data and figure supplement(s) for figure 2:

**Source data 1.** Full membrane scan for the western blot in *Figure 2B*.

**Figure supplement 1.** Deletion of Arginase 1 (*Arg1*) in myeloid cells decreases cell proliferation and cell death during early stages of pancreatic ductal adenocarcinoma.

**Figure supplement 2.** Deletion of Arginase 1 (*Arg1*) in myeloid cells alters the immune cell infiltration in a spontaneous pancreatic ductal adenocarcinoma mouse model during late stages of disease.

**Figure supplement 2—source data 1.** Single cell population levels used to calculate the percentage of each cell type in KF and KFCA.

**Figure supplement 3.** Deletion of Arginase 1 in myeloid cells decreases immune suppression in a spontaneous pancreatic ductal adenocarcinoma mouse model during late stages of the disease.

not statistically significant), accompanied by a reduction of desmoplastic stroma in KFCA, compared to age-matched KF pancreata (*Figure 2—figure supplement 1C*).

To investigate the causes underlying the trending reduction in PanIN formation, we performed an in-depth histological analysis. We previously showed that macrophages can directly promote epithelial cell proliferation during early stages in carcinogenesis (*Zhang et al., 2017b*). We and others have also demonstrated that macrophages also indirectly promote carcinogenesis by suppressing CD8$^+$ T cell infiltration and activation (*Bayne et al., 2012*; *Mitchem et al., 2013*; *Pylayeva-Gupta et al., 2012*; *Zhang et al., 2017a*; *Zhu et al., 2017*). Immunostaining for the proliferation marker Ki67 revealed a reduction in total cell proliferation, and a reduction in cell death as determined by cleaved caspase-3 (CC3) staining (*Figure 2—figure supplement 1D*). Immunofluorescence staining for immune cells by CD45 did not reveal changes in total CD45$^+$ cells (*Figure 2—figure supplement 1E*). Similarly, we did not find changes in infiltrating F4/80$^+$ macrophages or CD8$^+$ T cells by immunohistochemistry staining (*Figure 2—figure supplement 1F*). Thus, loss of *Arg1* in macrophages appears to reduce the ability of macrophages to promote proliferation but, at this stage, does not appear to correlate with changes in the immune system. These findings are consistent with our previous observations on the role of macrophages during the onset of carcinogenesis (*Zhang et al., 2017b*).

Given the trending reduction in PanIN formation, we next investigated the effects of myeloid *Arg1* deletion on progression to PDA. For this purpose, we aged KF and KFCA mice to 9–16 mo (*Figure 2D*), an age where we have previously observed invasive cancer formation in KF mice (*Garcia et al., 2020*). Accordingly, 16 out of 26 KF mice (62%) had invasive disease; in contrast, only 5 out of 24 KFCA mice had progressed (21%) (*Figure 2E*). We then stained the tissue for ARG1 together with F4/80, and CK19. As expected, we observed high ARG1 expression in macrophages in KF tissues and little to no ARG1 expression in macrophages in KFCA pancreata (*Figure 2F*). Surprisingly, we observed substantial ARG1 expression in epithelial cells from the KFCA pancreata, while epithelial cells in KF mice had little to no ARG1 expression (*Figure 2F* and *Figure 2—figure supplement 2A*). So, while deletion of *Arg1* in myeloid cells impairs malignant progression, it unleashes a compensatory upregulation of *Arg1* in epithelial cells.

We noted that the epithelia that expressed ARG1 in KFCA mice had an elongated appearance, consistent with tuft cells, a cell type that is not present in the healthy pancreas but common in low-grade PanINs (*Delgiorno et al., 2014*). By co-immunostaining for the tuft cell marker COX1, we confirmed that tuft cells are the exclusive source of epithelial ARG1 in these samples (*Figure 2—figure supplement 2A*), suggesting a tuft cell response to accumulating arginine in the microenvironment.

To comprehensively characterize the phenotype of myeloid *Arg1* deleted mice, we performed sc-RNA-seq on KF (n=1) and KFCA (n=1) pancreata dissected from 11 mo old mice. Unsupervised

clustering identified abundant stromal and immune cells, and a small population of epithelial cells, in both genotypes (*Figure 2G* and *Figure 2—figure supplement 2B and C*). The percentage of total macrophages was similar between KF and KFCA pancreas (*Figure 2—figure supplement 2D and E*, *Figure 2—figure supplement 3A and B*). We then subclustered the macrophages, from which we identified five different macrophage populations based on distinct gene profiles (*Figure 2H* and *Figure 2—figure supplement 3C*). The macrophage 1 population was defined by *Apoe*, *C1qa*, and *C1qc* markers of TAMs in mouse and human pancreatic cancer (*Kemp et al., 2021a*), while the macrophage 5 population expressed *Ear2* and *Retnla* (*Figure 2H*). Since macrophage APOE (Apolipoprotein E) is tumor promoting (*Kemp et al., 2021a*), we performed co-immunofluorescent staining for APOE, F4/80, and ECAD and detected a reduction in APOE expression (*Figure 2—figure supplement 3D and E*). scRNA-seq identified *Apoe* expression mainly in the KF macrophage 1 population with reduced expression in the KFCA population (*Figure 2—figure supplement 3F*).

Intriguingly, sc-RNA-seq revealed that deletion of *Arg1* in myeloid cells led to an upregulation of *Arg2* expression, mainly in the macrophage 4 population expressing *Chil3*, *Ly6c2*, and *Clec4e* genes (*Figure 2I*), suggesting the existence of a compensatory mechanism. We then performed reactome pathway enrichment analysis on the macrophage sc-RNA-seq data to determine both up- and down-regulated pathways (*Figure 2J* and *Figure 2—figure supplement 3G*). Interestingly, KFCA macrophages, mainly the macrophage 3 population, exhibited upregulation of signaling pathways involved in MHC I antigen processing and cross presentation (*Figure 2J* and *Figure 2—figure supplement 3H*), suggesting an improvement in antigen-specific CD8[+] T cell activation upon *Arg1* deletion. Among the differentially expressed genes in MHC I-mediated antigen processing and presentation were *H2-K1*, *Psmb8*, *H2-Q7*, *Ubb*, *H2-T23*, *Psmb9*, *H2-T22*, *Uba52*, *H2-Q6*, *Mrc1*, *Rbx1*, *Psmd8*, *Psmb5*, and *Znrf2* (*Figure 2—figure supplement 3H*).

## Deletion of Arginase 1 in myeloid cells increases CD8[+] T cell infiltration and activation

Macrophages in pancreatic cancer are highly immunosuppressive and have defective antigen presentation (*DeNardo and Ruffell, 2019*). As sc-RNA-seq suggested that *Arg1* null macrophages have higher antigen presenting activity, we stained KF and KFCA pancreata for CD8 and observed increased CD8[+] T cell infiltration in the latter (*Figure 3A*). We then sub-clustered T and natural killer (NK) cells in the sc-RNA-seq datasets from KF and KFCA pancreata and classified these into the following subclusters: naïve CD8[+] T cells, cytotoxic CD8[+] T cells, exhausted CD8[+] T cells, CD4[+] T cells, regulatory T cells (Treg), γδ T cells, NKT, and NK cells (*Figure 3B and C*). From this analysis, we observed an increase in the percentage of both cytotoxic and exhausted CD8[+] T cells in the KFCA model compared with KF (*Figure 3D and E*). Correspondingly, expression of genes related to CD8[+] T cell cytotoxicity, including *Granzyme b* (*Gzmb*), *Perforin 1* (*Prfn1*), and *Interferon gamma* (*Ifng*) was upregulated in CD8[+] T cells from the KFCA model (*Figure 3F*). Accordingly, co-immunofluorescent staining for CD8, GZMB, and ECAD showed an increase in GZMB-expressing CD8[+] T cells in KFCA pancreata (*Figure 3G* and H). We also found that the expression of genes involved in T cell exhaustion such as *cytotoxic T-lymphocyte associated protein 4* (*Ctla4*), *Furin*, *lymphocyte activating 3* (*Lag3*), and *programmed cell death 1* (*Pdcd1*) was upregulated in the KFCA model, compared with the KF (*Figure 3I*). Taken together, deletion of *Arg1* in myeloid cells resulted in an increase in CD8[+] T cell infiltration and activation, counterbalanced by an increase in exhaustion as well.

## Systemic arginase inhibition in combination with anti-PD1 immune checkpoint reduces tumor growth

Our genetic model revealed that deletion of *Arg1* in myeloid cells reduced PanIN/PDA progression and was accompanied by an increased infiltration of CD8[+] T cells. However, we also observed several compensatory mechanisms that may have blunted the effect of *Arg1* loss. These included expression of *Arg2* in myeloid cells and expression of *Arg1* in epithelial cells. We reasoned that systemic inhibition of arginase (*Steggerda et al., 2017*), using a pharmacologic approach might bypass these compensatory mechanisms.

We implanted a KPC pancreatic cancer cell line (7940B) (*Long et al., 2016*) orthotopically into the pancreas of syngeneic C57BL6/J mice. Upon tumor detection (by palpation or by ultrasound), we randomly divided the mice into two groups to receive either vehicle or Arginase inhibitor (CB-1158,

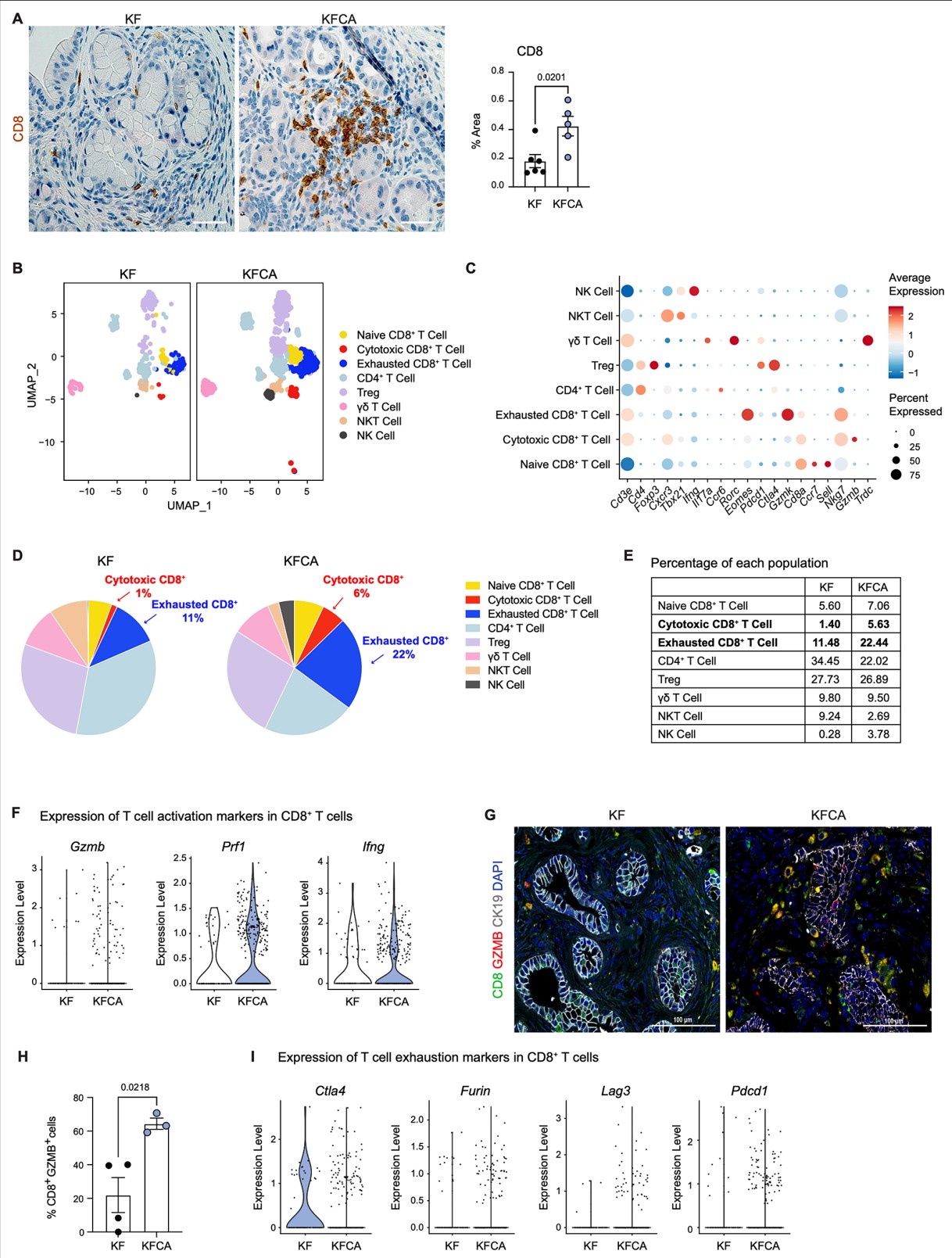

**Figure 3.** Arginase 1 deletion in myeloid cells increases CD8⁺ T cell infiltration and activation in a spontaneous pancreatic ductal adenocarcinoma mouse model. (**A**) Representative images of CD8 immunohistochemistry staining (brown) in KF and KFCA tissue. Scale bar, 50 μm. Quantification of positive area on the right, n=5–6/group. Student's t test was used to determine statistical significance. (**B**) Uniform manifold approximation and projection (UMAP) visualization of defined T and natural killer (NK) cell clusters comparing single-cell RNA sequencing (sc-RNA-seq) data from KF and

*Figure 3 continued on next page*

*Figure 3 continued*

KFCA. (**C**) Dot plot of lineage markers used to identify the different types of lymphocytes. Dot size shows expression frequency, dot color shows average expression. (**D**) Pie charts showing the proportion of the identified lymphocyte populations in KF and KFCA sc-RNA-seq, percentage values are provided for populations that differ dramatically between KF and KFCA. (**E**) Table showing the percentage of each identified lymphocyte population in KF and KFCA sc-RNA-seq. (**F**) Violin plots showing normalized expression levels of T cell activation markers in all CD8+ T cell populations identified in KF and KFCA sc-RNA-seq data. (**G**) Representative images of co-immunofluorescence staining for CD8 (green), GZMB (red), CK19 (gray), and DAPI (blue). Scale bar, 100 µm. (**H**) Quantification of CD8+ GZMB+ cells, n=3/group. Student's t test was used to determine significance. (**I**) Violin plots showing normalized expression levels of T cell exhaustion markers in all CD8+ T cell populations identified in KF and KFCA.

The online version of this article includes the following source data for figure 3:

**Source data 1.** Single cell population levels used to calculate the percentage of lymphocytes in KF and KFCA.

Calithera Biosciences) (*Figure 4A*). We pharmacologically treated the mice for 10 d and then harvested the tumors 20 d post-implantation. First, we investigated the infiltration of CD8+ T cells by staining the vehicle and CB-1158 mouse tissue for CD8 by immunohistochemistry . Interestingly, we observed that systemic inhibition of Arginase by CB-1158 increased the infiltration of CD8+ T cells (*Figure 4B and C*), recapitulating our findings from the genetically engineered model. To identify whether there was an increase in CD8+ T cell activation, we stained the tissue samples for CD8 and the activation marker GZMB, as well as the epithelial marker ECAD. While in the control tissue, GZMB expression was rare, in the CB-1158 treatment group, GZMB was common in CD8+ T cells (*Figure 4D and E*).

Data from the genetic model showed an increase in T cell activation but also an increase in exhaustion. We thus repeated the syngeneic orthotopic transplantation experiment described above, but this time we added blockade of the PD1 immune checkpoint to circumvent T cell exhaustion. Tumor bearing mice were divided into four different groups to receive: (1) vehicle + IgG control, (2) arginase inhibitor (CB-1158) + IgG, (3) vehicle + anti-PD1, and (4) CB-1158 + anti-PD1 (*Figure 4F*). We observed a trending decrease in tumor weight in the CB-1158 + IgG group and a significant decrease in tumor weight in the combination group of CB-1158 + anti-PD1 immune checkpoint blockade compared with our vehicle control group (*Figure 4G*). These findings recapitulated the decrease in tumor formation observed in the KFCA mouse model.

We then proceeded to characterize the tumor tissue. We examined ARG1 expression in macrophages (F4/80+) and epithelial (ECAD+) cells by co-immunofluorescence staining and observed no changes in ARG1 distribution or expression among treatment groups (*Figure 4—figure supplement 1A*), consistent with the notion that the CB-1158 Arginase inhibitor does not affect enzyme production but suppresses its activity. Hematoxylin and eosin (H&E) staining showed large necrotic areas in both CB-1158 and CB-1158 + anti-PD1 groups (*Figure 4—figure supplement 1B*). To further characterize the tumor tissue, we stained for cell proliferation (Ki67) and apoptotic cell death (CC3) by immunohistochemistry staining (*Figure 4—figure supplement 1C and D*). The combination treatment group showed a significant decrease in cell proliferation compared to the vehicle control group (*Figure 4—figure supplement 1C and E*). We also observed either a significant or trending increase in CC3 upon CB-1158 + IgG, vehicle + anti-PD1, or the combination treatment (*Figure 4—figure supplement 1D and F*). Additionally, metabolomics analysis of these orthotopic tumors revealed an increase in arginine levels in the tumors from mice treated with CB-1158 and anti-PD1 (*Figure 4H*).

Arginase breaks down arginine into urea and ornithine. Thus, it is equally conceivable that lack of arginine (Arg) or excess of ornithine (Orn) and/or urea might impair the ability of CD8+ T cells to proliferate and/or become activated. To determine how loss of Arginase increases CD8+ T cell proliferation and activation, we performed an in vitro CD8+ T cell proliferation and activation assay. We isolated naïve CD8+ T cells from spleen and lymph nodes of C57BL/6 mice and cultured them for 72 hr with complete RPMI media or with media lacking arginine (−Arg RPMI). In both media conditions, we added groups where the media was supplemented with either ornithine or urea, or with a combination of both (*Figure 4I*). We found that arginine deprivation results in reduced CD8+ T cell proliferation and activation (*Figure 4J and K*). Additionally, we found that culturing cells in RPMI media containing urea sightly enhanced CD8+ T cell proliferation and activation, compared to RPMI media control (*Figure 4J and K*, gray bars), while no enhancement was observed with ornithine supplementation.

When cells were cultured in RPMI media lacking arginine, supplementation with ornithine, but not urea, partially rescued the proliferation defect caused by lack of arginine. No further rescue occurred adding urea; neither ornithine nor urea rescued the activation defect caused by lack of arginine. We

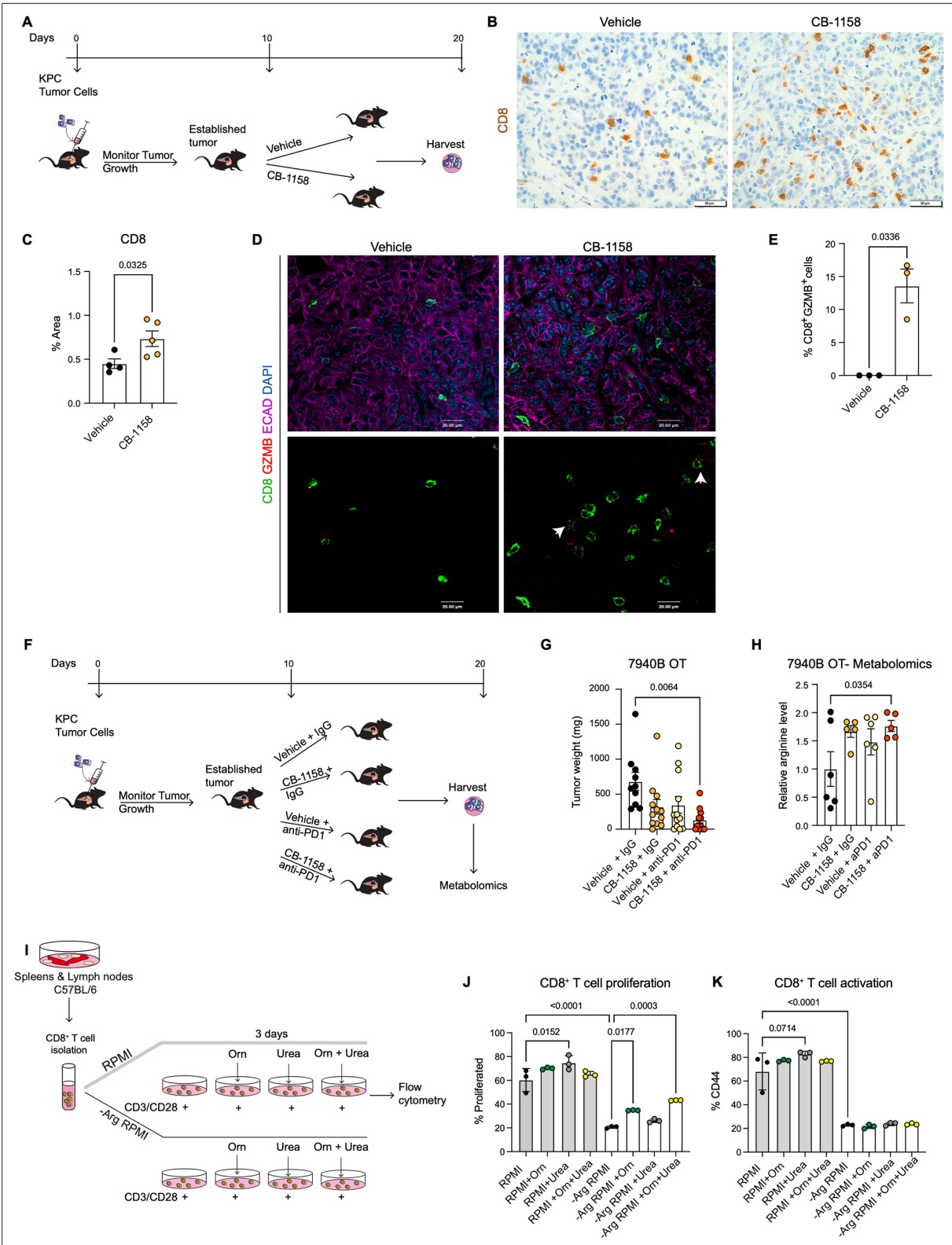

**Figure 4.** Systemic inhibition of arginase by CB-1158 in combination with anti-PD1 reduces tumor growth in an orthotopic pancreatic ductal adenocarcinoma mouse model. (**A**) Experimental timeline and design for orthotopic transplantation of 7940B KPC cells in syngeneic mice. (**B**) Representative images of immunohistochemistry staining for CD8 (brown). Scale bar, 50 μm. (**C**) Quantification of positive area of CD8 staining from B. Statistical significance was determined using unpaired t-test with Welch's correction. (**D**) Representative images of co-immunofluorescence staining for CD8 (green), GZMB (red), ECAD (purple), and DAPI (blue) in the vehicle and CB-1158 treated group. Scale bar, 20 μm. White arrows point to co-

*Figure 4 continued on next page*

*Figure 4 continued*

localization of CD8 (green) with GZMB (red). (**E**) Percentage of CD8⁺ and GZMB⁺ cells. Statistical significance was determined using unpaired t-test with Welch's correction. (**F**) Experimental timeline and design for orthotopic transplantation of 7940B KPC cells in syngeneic mice and different treatment groups. (**G**) Final tumor weight (mg) from the different treatment groups, n=10–12/group. Statistical significance was determined using two-way ANOVA with Tukey's multiple comparisons correction test. (**H**) Metabolomic analysis showing relative arginine levels in the tumors of the different treatment groups. Two-way ANOVA was used to determine statistical significance, n=5–6/group. (**I**) Experimental scheme for CD8⁺ T cell isolation and growth in different media conditions. −Arg RPMI = RPMI lacking arginine, Orn = ornithine. (**J**) Percent of proliferated CD8⁺ T cells from flow cytometry analysis. (**K**) Percent of activated CD8⁺ T cells (CD44) from flow cytometry analysis. Two-way ANOVA with Tukey's multiple comparisons correction test was used to determine significance. n=3/condition.

The online version of this article includes the following figure supplement(s) for figure 4:

**Figure supplement 1.** Histological analysis of the multiple treatment groups in the orthotopic mouse model.

**Figure supplement 2.** Flow cytometry gating strategy for CD8⁺ T cell proliferation and activation.

thus concluded that CD8⁺ T cells require arginine for proliferation and activation, and that the arginine breakdown byproducts are not harmful to CD8⁺ T cell proliferation nor are they able to fully rescue lack of arginine.

Together, these findings indicate that Arginase inhibition in combination with anti-PD1 reduces immune suppression and decreases pancreatic cancer tumor growth (see working model in *Figure 5*).

## Discussion

Abundant myeloid cells in the TME are associated with poor prognosis in multiple types of cancer, including PDA (*Gentles et al., 2015*; *Sanford et al., 2013*; *Tsujikawa et al., 2017*). In contrast, high levels of T cells correlate with longer survival (*Gentles et al., 2015*; *Balachandran et al., 2017*). We and others have shown that myeloid cells promote pancreatic cancer growth both directly and by inhibiting CD8⁺ T cell anti-tumor immunity (*Zhang et al., 2017b*; *Zhang et al., 2017a*; *Mitchem et al., 2013*; *Halbrook et al., 2019*; *Steele et al., 2016*; *Liou et al., 2015*). The specific mechanisms by which macrophages drive immune suppression, and how to best target macrophages to improve outcomes in pancreatic cancer remain poorly understood.

Macrophages are plastic cell types, traditionally classified into pro-inflammatory 'M1' and anti-inflammatory 'M2' subtypes, based on their gene expression pattern (*Mantovani et al., 2002*). Recently, a wealth of evidence now supports the notion that TAMs are distinct from M1 and M2 (*Boyer et al., 2022*; *DeNardo and Ruffell, 2019*; *Halbrook et al., 2019*). Adding further complexity, TAMs also exhibit heterogeneous populations within an individual tumor. ARG1 is classically considered a

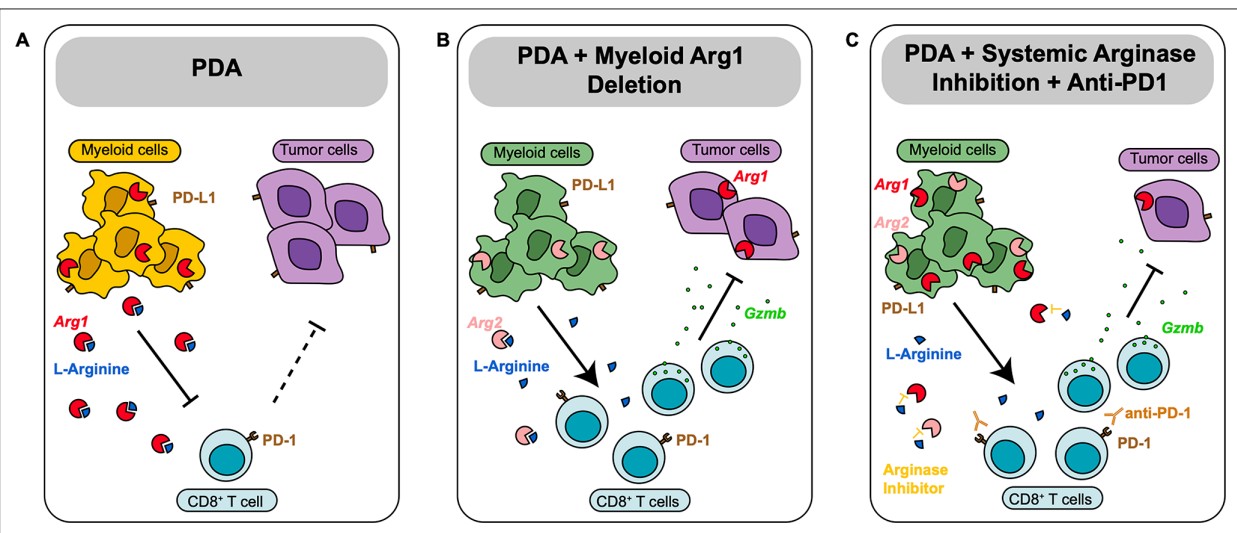

**Figure 5.** Diagram depicting our working model. (**A**) In pancreatic ductal adenocarcinoma (PDA), expression of *Arg1* in myeloid cells is immune suppressive. (**B**) Deletion of *Arg1* in myeloid cells in a spontaneous PDA mouse model induces macrophage repolarization and decreases tumor formation. (**C**) Systemic Arginase inhibition in combination with anti-PD1 immune checkpoint blockade further decreases tumor growth.

'marker' of the anti-inflammatory M2 state. In this study, we show that Arg1 is expressed in human and mouse TAMs, as well as in other myeloid populations. Furthermore, when we stratified human PDA based on *ARG1* expression, we found inverse correlations between *ARG1* and survival. However, prior to this work, a functional role of Arg1 in pancreatic TAMs and tumor immunity had not been evaluated.

Arginases are enzymes that hydrolyze the amino acid L-arginine to urea and L-ornithine in the liver urea cycle (*Jenkinson et al., 1996*). An analysis of plasma metabolites and tumor interstitial fluid in an autochthonous PDA mouse model revealed that L-arginine was drastically reduced in the tumor interstitial fluid (*Sullivan et al., 2019*). Additionally, ornithine levels increased in the PDA tumor interstitial fluid compared to plasma (*Sullivan et al., 2019*), suggesting arginase contribution to immunosuppression through depletion of arginine.

There are two isoforms of arginase, ARG1 and ARG2, located in the cytoplasm and mitochondria, respectively (*Gotoh et al., 1996*; *Munder et al., 2005*). The arginase genes share a 58% sequence identity (*Haraguchi et al., 1987*) and are almost identical at the catalytic site. *Arg1* is normally expressed by hepatocytes, but it is also expressed in TAMs in a variety of tumor types. *Arg2* is located in various cell types, including renal cells, neurons, and macrophages (*Bronte and Zanovello, 2005*; *Caldwell et al., 2018*). In a mouse model of Lewis lung carcinoma, myeloid cells express high levels of *Arg1*, resulting in impaired T cell function (*Rodriguez et al., 2004*). In that study, 3LL tumor cells were implanted subcutaneously in the right flank of the mice, and at the same time, mice were treated with the pan-arginase inhibitor tool compound N-hydroxy-nor-L-Arg. Arginase inhibition reduced subcutaneous tumor growth in a dose-dependent manner. The anti-tumor effect of the arginase inhibitor was in part dependent on T cell function, as inhibition on tumor growth was not observed when mice lacking a functional immune system were treated with the arginase inhibitor (*Rodriguez et al., 2004*). In another study (*Miret et al., 2019*), inhibition of arginase activity by the arginase inhibitor tool compound Cpd9 (*Van Zandt et al., 2013*) decreased T cell suppression and reduced tumor growth in a Kras[G12D] lung cancer mouse model. In a more recent study using a mouse model of neuroblastoma, myeloid *Arg1* expression promoted tumor growth (*Van de Velde et al., 2021*).

To address the role of macrophage ARG1 in PDA, we used a dual-recombinase genetic approach to delete *Arg1* in myeloid cells and aged the mice until most animals in the control group developed invasive PDA. We discovered that deletion of *Arg1* in myeloid cells reduced progression to invasive disease, conversely resulting in accumulation of early lesions with prominent Tuft cells, a cell type that is protective toward tumor progression (*DelGiorno et al., 2020*; *Hoffman et al., 2021*). Reduced tumor progression was accompanied by changes in the immune microenvironment, such as an increase in infiltration and activation of CD8[+] T cells. However, a fraction of the mice developed invasive disease. Complete CD8[+] T cell reactivation against tumors did not occur as we also observed an increase in exhausted CD8[+] T cells. Additionally, from sc-RNA-seq, we observed increased *Arg2* expression in myeloid cells. This finding is in line with a report in which patients with ARG1 deficiency had an increase in ARG2 levels (*Crombez and Cederbaum, 2005*; *Grody et al., 1993*), suggesting compensation of ARG2 for the loss of ARG1.

Further analysis revealed that, in the absence of myeloid *Arg1*, the chemosensory tuft cells began to express ARG1. Tuft cells sense their surrounding environment and respond in a variety of ways, including modifying immune responses in a way that restrains PDA malignant progression (*DelGiorno et al., 2020*; *Hoffman et al., 2021*). Their compensation for the loss of myeloid *Arg1* expression in what is likely now an arginine-rich microenvironment is consistent with these functions and suggests that systemic inhibition of ARG1 may have a more profound effect than its ablation from myeloid cells alone.

To systemically inhibit ARG1, we used CB-1158 (INCB001158), an orally bio-available small molecule inhibitor of arginase. CB-1158 is not cell permeable and thus does not inhibit liver ARG1, which would lead to immediate toxicity (*Steggerda et al., 2017*). In vitro, CB-1158 inhibits human recombinant ARG1 and ARG2. The in vivo action is attributed to inhibition of extracellular ARG1 released by TAMs and other myeloid cells (*Steggerda et al., 2017*). We used a syngeneic, orthotopic transplantation model based on KPC pancreatic cancer cells transplanted in C57Bl6/J mice (*Hingorani et al., 2005*; *Long et al., 2016*). Treatment with CB-1158 led to an increase in infiltrating CD8[+] T cells, recapitulating the findings in the spontaneous model. Immune checkpoint therapies such as PD-1/PD-L1 used to enhance the anti-tumor immune response of T cells are not effective in pancreatic cancer (*Brahmer et al., 2012*), in part due to resistance mechanisms induced by myeloid cells.

We found that the combination treatment of CB-1158 with anti-PD1 immune checkpoint blockade reactivated exhausted CD8[+] T cells and decreased tumor growth. CB-1158 is currently evaluated as a single agent and in combination with immune checkpoint therapy or chemotherapy in patients with solid tumors (https://www.clinicaltrials.gov/, NCT02903914 and NCT03314935). An open question is whether the inhibitor also has a direct effect on epithelial cells. In a model of obesity-driven pancreatic cancer, *Arg2* is upregulated in epithelial cells, and its knockdown reduces tumor growth in a cell autonomous manner, independently from immune responses (*Zaytouni et al., 2017*). While we did not observe *Arg2* expression in epithelial cells in non-obese KF mice, analysis of human data revealed low, but detectable expression of *Arg2*, possibly supporting the possibility that, in human patients, CB-1158 might work through systemic inhibition of both arginase isoforms in multiple cell compartments, including epithelial cells.

Because of the heterogeneity of myeloid cells acquired by their interaction with the TME, myeloid cells most likely utilize multiple mechanisms that affect their function. Our findings show that one of the mechanisms by which myeloid cells promote immune suppression and tumor growth in pancreatic cancer is through overexpression and activity of *Arg1*. Thus, arginase inhibition may be an effective therapeutic strategy to enhance anti-tumor immune responses.

# Materials and methods

## Key resources table

| Reagent type (species) or resource | Designation | Source or reference | Identifiers | Additional information |
|---|---|---|---|---|
| Strain, strain background (*Mus musculus*) | C57BL/6 J | Jackson Laboratory | Stock #: 000664 | |
| Strain, strain background (*M. musculus*) | *Arg1^{f/f}* | Jackson Laboratory | Stock #: 008817 | |
| Strain, strain background (*M. musculus*) | KF | *Wen et al., 2019* | | |
| Strain, strain background (*M. musculus*) | KFCA | This paper | | Pasca di Magliano Lab, University of Michigan |
| Cell line (*M. musculus*) | 7940B | *Long et al., 2016* | | KPC cell line |
| Antibody | ARG1 (Rabbit, monoclonal) | Cell Signaling, Danvers, MA | Cat #: 93668 S | IF: 1:75 WB: 1:1000 |
| Antibody | CD45 (Mouse, monoclonal) | R&D Systems, Minneapolis, MN | Cat #: MAB14302 | IF: 1:400 |
| Antibody | ECAD (Mouse, monoclonal) | Cell Signaling, Danvers, MA | Cat #: 14472 S | IF: 1:50 |
| Antibody | F4/80 (Rabbit, monoclonal) | Cell Signaling, Danvers, MA | Cat #: 70076 S | IF: 1:250 IHC: 1:250 |
| Antibody | CK19, Troma III (Rat, monoclonal) | Developmental Studies Hybridoma Bank, Iowa City, IA | | IF: 1:50 |
| Antibody | Ki67 (Rabbit, polyclonal) | Abcam, Cambridge, UK | Cat #: ab15580 | IHC: 1:1000 |
| Antibody | CC3 (Rabbit, polyclonal) | Cell Signaling, Danvers, MA | Cat #: 9661 L | IHC: 1:300 |
| Antibody | CD8 (Rabbit, monoclonal) | Cell Signaling, Danvers, MA | Cat #: 98941 S | IHC: 1:300 IF: 1:400 |
| Antibody | APOE (Rabbit, monoclonal) | Abcam | Cat #: ab183597 | IF: 1:500 |
| Antibody | COX1 (Goat, polyclonal) | Santa Cruz | Cat #: Sc-1754 | IF: 1:200 |
| Antibody | GZMB (Rabbit, monoclonal) | Cell signaling | Cat #: D2H2F | IF: 1:800 |
| Antibody | Vinculin (Rabbit, monoclonal) | Cell signaling | Cat #: 13901 S | WB: 1:1000 |
| Commercial assay, kit | Alexa fluor 488 Tyramide SuperBoost kit | Invitrogen | Cat #: B40922 | |

*Continued on next page*

*Continued*

| Reagent type (species) or resource | Designation | Source or reference | Identifiers | Additional information |
|---|---|---|---|---|
| Commercial assay, kit | RNA Scope Multiplex Fluorescent v2 Detection Kit | Advanced Cell Diagnostics | Cat #: 323110 | |
| Commercial assay, kit | RNAscope Probe-Hs-ARG1 | Advanced Cell Diagnostics | Cat #: 401581 | |
| Chemical compound, drug | CB-1158 | Calithera Biosciences, Inc, South San Francisco, CA | | 100 mg/kg, Oral gavage, twice a day |
| Chemical compound, drug | Anti-PD1 | BioXcell | Cat #: BE0033-2, clone J43 | 200 µg/i.p. Every 3 d |
| Software, algorithm | R Studio | Rstudio.com | Version: 4.1.1 -- "Kick Things" | |
| Software, algorithm | ImageJ | Imagej.nih.gov | Version: 2.0.0-rc-69/1.52 p | |
| Software, algorithm | Adobe Illustrator | Adobe.com | 2022 | |
| Software, algorithm | Halo software | Indica Labs | | |
| Software, algorithm | Prism 9 for macOS | Graphpad.com | Version: 9.4.1 (458), July 18, 2022 | |

## Mice studies

*Arg1^{f/f}* mice (Stock # 008817) and C57BL/6 J WT mice (Stock# 000664) were obtained from the Jackson Laboratory and bred in-house. *Arg1^{f/f}* mice were generated to have *loxP* sites flanking exons 7 and 8 in the *Arg1* gene (**El Kasmi et al., 2008**). *Lyz2^{Cre/+}* mice and KF (*Kras^{Frt-STOP-Frt-G12D/+};Ptf1a^{FlpO/+}*) mice were donated by Dr. Howard Crawford. *Lyz2^{Cre/+}* mice express Cre in myeloid cells due to the insertion of the Cre cDNA into the endogenous M lysozyme (LysM) locus (**Clausen et al., 1999**). *Lyz2^{Cre/+};Arg1^{f/f}* mice were generated by crossing *Lyz2^{Cre/+}* mice with *Arg1^{f/f}* mice. KFCA mice were generated by crossing KF mice with *Lyz2^{Cre/+};Arg1^{f/f}* mice. Age matched male and female mice were used in these studies.

## Orthotopic surgery procedure

50 µl of 50,000 7940B cells (C57BL/6 J) resuspended in a 1:1 ratio of RPMI medium 1640 (Gibco, 11875093) and Matrigel matrix basement membrane (Corning, 354234) were injected into the pancreas of C57BL/6 J mice. For surgery procedure, see **Aiello et al., 2016**. 7940B cells were derived from a male KPC (*Kras^{LSL-G12D/+};p53^{LSL-R172H/+};Ptf1a^{Cre/+}*) mouse tumor (**Long et al., 2016**). Cells were tested for mycoplasma by MycoAlertTM Plus Mycoplasma Detection Kit (Lonza). Once mice had established tumors, around day 10 after implantation, mice were randomized into different treatment groups.

## Chemical compounds

Arginase inhibitor, CB-1158 was synthesized and provided by Calithera Biosciences, Inc, South San Francisco, CA (2017). For the mice studies, CB-1158 was dissolved in Milli-Q water and administered by oral gavage twice a day at 100 mg/kg. This treatment started 10 d after tumor implantation and lasted for 10 d. The control group received Milli-Q water by oral gavage, twice a day. Purified anti-mouse PD1 antibody (BioXcell #BE0033-2, clone J43) was used for the in-vivo anti-PD1 blockade experiments. Anti-PD1 was used at a dose of 200 µg/i.p. injection, every 3 d. The control group received Polyclonal Armenian hamster IgG (BioXcell, BE0091), and it was administered in parallel to anti-PD1.

## Single-cell RNA-seq

Human sc-RNA-seq data were previously published in **Steele et al., 2020** (NIH dbGaP database accession #phs002071.v1.p1). Healthy mouse pancreas sc-RNA-seq data were previously published in **Kemp et al., 2021b** (NIH dbGaP database accession #GSM5011581), and mouse spontaneous PDA sc-RNA-seq data generated using the KPC (*Kras^{LSL-G12D/+};p53^{LSL-R172H/+};Ptf1a^{Cre/+}*) model was previously published (NIH dbGap databse accession GSE202651). To generate the KF and KFCA

sc-RNA-seq data, pancreatic tissue was harvested from KF (n=1) and KFCA mice (n=1) at 11 mo of age. The tissue was mechanically minced, then digested with Collagenase V (Sigma C9263, 1 mg/ml in RPMI) for 30 min at 37°C with shaking. Digestions were filtered through 500 µm, 100 µm, and 40 µm mesh to obtain single cells. Dead cells were removed using the MACS Dead Cell Removal Kit (Miltenyi Biotec). Single-cell complementary DNA libraries were prepared and sequenced at the University of Michigan Advanced Genomics Core using the 10× Genomics Platform. Samples were run using 50-cycle paired-end reads on the NovaSeq 6000 (Illumina) to a depth of 100,000 reads. The raw data were processed and aligned by the University of Michigan Advanced Genomics Core. Cell Ranger count version 4.0.0 was used with default settings, with an initial expected cell count of 10,000. Downstream sc-RNA-seq analysis was performed using R version 4.0.3, R package Seurat version 4.0.2, and R package SeuratObject version 4.0.1 (RStudio Team RStudio: Integrated Development for R [RStudio, 2015]); http://www.rstudio.com/ R Core Development Team R: A Language and Environment for Statistical Computing (R Foundation for Statistical Computing, 2017); https://www.R-project.org/ (*Butler et al., 2018*; *Stuart et al., 2019*). Data were filtered to only include cells with at least 100 genes and genes that appeared in more than 3 cells. Data were normalized using the NormalizeData function with a scale factor of 10,000 and the LogNormalize normalization method. Data were then manually filtered to exclude cells with <1000 or >60,000 transcripts and <15% mitochondrial genes. Variable genes were identified using the FindVariableFeatures function. Data were scaled and centered using linear regression of transcript counts. PCA was run with the RunPCA function using the previously defined variable genes. Cell clusters were identified via the FindNeighbors and FindClusters functions, using dimensions corresponding to approximately 90% variance as defined by PCA. UMAP clustering algorithms were performed with RunUMAP. Clusters were defined by user-defined criteria. The complete R script including figure-specific visualization methods is publicly available on GitHub (https://github.com/PascaDiMagliano-Lab/Arginase-1-is-a-key-driver-of-immune-suppression-in-pancreatic-cancer, copy archived at swh:1:rev:ddd8595b-93230f76ebc219588953cd36f4941572; *Donahue, 2023*).

## Kaplan-Meier survival analysis

For Kaplan-Meier overall survival, we used the human dataset GSE71729 containing 125 primary PDA tumor samples. The samples were split into *ARG1*-low (n=62) and *ARG1*-high (n=63) groups. Survival analysis with log-ranked test was subsequently plotted in GraphPad Prism v9.

## Histopathology

Tissues were fixed in 10% neutral-buffered formalin overnight, dehydrated, paraffin-embedded, and sectioned into slides. H&E and Gomori's Trichrome staining were performed according to the manufacturer's guidelines.

## Immunohistochemistry (IHC)

Paraffin sections were re-hydrated with two series of xylene, two series of 100% ethanol, and two series of 95% ethanol. Slides were rinsed with water to remove previous residues. CITRA Plus (BioGenex) was used for antigen retrieval and microwaved for 5 min and then 3 min. Once cool down, sections were blocked with 1% bovine serum albumin (BSA) in PBS for 30 min. Primary antibodies were used at their corresponding dilutions (Key resources table) and incubated at 4°C overnight. Biotinylated secondary antibodies were used at a 1:300 dilution and applied to sections for 45 minat room temperature (RT). Sections were then incubated for 30 min with ABC reagent from Vectastain Elite ABC Kit (Peroxidase), followed by DAB (Vector).

## Co-immunofluorescence (Co-IF)

Deparaffinized slides were blocked with 1% BSA in PBS for 1 hr at RT. Primary antibodies (Key resources table) were diluted in blocking buffer and incubated overnight at 4°C, followed by secondary antibody (Alexa Fluor secondaries, 1:300) for 45 min at RT. Slides were mounted with Prolong Diamond Antifade Mountant with DAPI (Invitrogen). TSA Plus Fluorescein (PerkinElmer) was also used in the Co-IF for primary antibodies.

## In situ hybridization (ISH) with Co-IF

The RNA Scope Multiplex Fluorescent Detection Kit (Advanced Cell Diagnostics) was used according to the manufacturer's protocol. The probe used for *ARG1* was Hs-ARG1 (401581, Advanced Cell Diagnostics). Freshly cut human paraffin-embedded sections were baked for 1 hr at 60°C prior to staining. Slides were then deparaffinized and treated with hydrogen peroxide for 10 min at RT. Target retrieval was performed in a water steamer boiling for 15 min, and then slides were treated with the ProteasePlus Reagent (Advanced Cell Diagnostics) for 30 min. The RNA scope probe was hybridized for 2 hr at 40°C. The signal was amplified using the AMP (amplification) materials provided in the ACD Multiplex Kit (Advanced Cell Diagnostics). The signal was developed with horseradish peroxidase (HRP) channel. Once completed, the samples were washed in PBS, and then blocked for 1 hr with 5% donkey serum at RT. Primary antibody against CD45 (1:400) was incubated overnight at 4°C. Secondary antibodies (1:300 in blocking buffer) were incubated for 1 hr at RT, and samples were washed three times in PBS. Slides were counterstained with DAPI and mounted with ProLong Gold Antifade Mountant (Thermo Fisher Scientific).

Images were taken either with an Olympus BX53 microscope, a Leica SP5 microscope, a Leica STELLARIS 8 FALCON Confocal Microscopy System, or scanned with a Pannoramic SCAN scanner (Perkin Elmer). Quantification of positive cell number or area was done using ImageJ, three to five images/slide (200× or 400× magnification) taken from three to four samples per group or using the Halo software (Indica Labs).

## Cell culture

The 7940B cells were cultured in Dulbecco's Modified Eagle Medium (DMEM, 11965–092) supplemented with 10% fetal bovine serum (FBS) and 1% penicillin streptomycin. Tumor cell CM were collected from the 7940B cells that were cultured to confluency. Media were centrifuged at 300 *g* for 10 min at 4°C to remove contaminating tumor cells. These CM were used for the macrophage polarization assay. For macrophage polarization, BM cells were isolated from WT or *Lyz2*$^{Cre/+}$*;Arg1*$^{f/f}$ mice femurs. Once isolated, BM cells were cultured with 7940B cell CM plus total DMEM media at a 1:1 ratio for 6 d. Fresh media were added during day 3. This process allowed the differentiation and polarization of BM cells to TAMs (*Zhang et al., 2017a*).

## In vitro CD8 T cell culture

Naïve CD8 T cells were isolated from C57BL/6 mouse spleens and lymph nodes by magnetic bead separation (Miltenyi Biotec, cat # 130-096-543), following the manufacturers' protocols. T cells were activated with plate-bound anti-CD3e (2 µg/ml) and soluble anti-CD28 (1 µg/ml) in the presence of mIL-2 (10 ng/ml; R&D Systems). Cells were cultured in media constituted with RPMI powder (USBiological, R9010-01) according to the manufacturer's instructions and supplemented with 10% FBS and 50 µM 2-Mercaptoethanol (Gibco) or in RPMI medium lacking arginine (−Arg RPMI). 1.15 mM arginine was added to the RPMI medium. Ornithine (Orn) and urea were added at 1.15 mM to the corresponding media. $2 \times 10^5$ cells in 200 µL of media were plated in a flat-bottom 96-well plate and cultured for 3 d. To assess proliferation, cells were labeled with 5 µM Cell Trace Violet in 0.1% BSA/PBS for 30 min at 37°C prior to activation.

## Western blot

TAMs from WT or *Lyz2*$^{Cre/+}$*;Arg1*$^{f/f}$ mice were lysed in RIPA buffer (Sigma-Aldrich) with protease and phosphatase inhibitors (Sigma-Aldrich). Protein samples were quantified, normalized, and then electrophoresed in a 4–15% SDS-PAGE gel (BioRad). Protein was transferred to a PVDF (polyvinylidene difluoride) membrane (BioRad), blocked with 5% milk for 1 hr at RT, and then incubated with primary antibodies overnight at 4°C (Key resources table). Membranes were then incubated with HRP-conjugated secondary antibody (1:5000) for 2 hr at RT. Membranes were washed, incubated in Western Lightning Plus-ECL (PerkinElmer), and then visualized with the ChemiDoc Imaging System (BioRad).

## Metabolomics analysis

CM (200 µL) was collected from each well of WT and *Lyz2*$^{Cre/+}$*;Arg1*$^{f/f}$ TAMs in a six-well plate after 6 d of culture and used for extracellular metabolite profiling. Briefly, to the 200 µL of media, 800 µL of

ice-cold 100% methanol was added. Cell lysates from parallel plates were used for protein quantification, and the protein amount was used to normalize the volume of samples collected for metabolomics. The samples were centrifuged at 12,000 *g* for 10 min after which the supernatant was collected, dried using SpeedVac Concentrator, reconstituted with 50% v/v methanol in water, and analyzed by targeted LC-MS/MS and processed as previously described (*Nelson et al., 2020*). For metabolomics in orthotopic tumors, ~50 mg of tumor tissue was grinded into powder with mortar and pestle containing liquid nitrogen. Thereafter, the samples were suspended in 1 mL of 80% methanol in a 1.5-mL Eppendorf tube and stored overnight at −80°C. Next day, samples were centrifuged at top speed, and supernatant was collected into a new 1.5-mL Eppendorf tube, dried, and processed for LC-MS/MS as with the in vitro samples.

## Statistics

GraphPad Prism 9 was used to perform statistical analysis. T-test or ANOVA was performed for group comparisons. R software 3.5.2 was used for the analysis of the microarray data set. A p-value was considered statistically significant when p<0.05.

## Acknowledgements

We thank Daniel Long and Michael Mattea for histology services at the University of Michigan. We thank Calithera Biosciences for providing the Arginase inhibitor, CB-1158. We thank the Advanced Genomics core at the University of Michigan for RNA sequencing. We also thank the Microscopy Core and the Flow Cytometry Core at the University of Michigan Biomedical Research Core Facilities for providing access to advanced microscopy and flow cytometry. We thank the Microscopy, Imaging and Cellular Physiology Core of the Michigan Diabetes Research Center at the University of Michigan for providing access to the Leica STELLARIS 8 FALCON Confocal Microscopy System, funded by grant NIH S10OD28612-01-A1. This study was supported by NIH/NCI U01CA224145, R01CA151588, R01CA198074, R01CA268426, U01CA274154, by an American Cancer Society Scholar grant and by the Pancreatic Cancer Action Network to M Pasca di Magliano. This study was also supported by the University of Michigan Cancer Center Support Grant (NCI P30CA046592), including an Administrative Supplement to M Pasca di Magliano. CA Lyssiotis was supported by the NCI (R37CA237421, R01CA248160, R01CA244931). HC Crawford was supported by R01 CA247516. RE Menjivar was supported by the University of Michigan Rackham Merit Fellowship, by the Cellular and Molecular Biology Training Grant (NIH T32-GM007315), by the Center for Organogenesis Training Program (NIH T32-HD007505), and by the NCI (F31-CA257533). ZC Nwosu was supported by the Michigan Postdoctoral Pioneer Program, University of Michigan Medical School. WD was supported by University of Michigan Training Program in Organogenesis. KL Donahue was supported by the Cancer Biology Program training grant (T32-CA009676). HSH was supported by 2T32AI007413 and T32DK094775. A Velez-Delgado was supported by the Rackham Merit Fellowship, by the Cellular Biotechnology Training Program (T32GM008353) and by the NCI (F31-CA247037). P Kadiyala was supported by the Immunology training grant (T32-AI007413). D Salas-Escabillas was supported by the Rackham Merit Fellowship and by the Cancer Biology Program training grant (T32-CA009676). ES Carpenter was supported by the American College of Gastroenterology Clinical Research Award and by T32-DK094775. Y Zhang was funded by the NCI-R50CA232985. CJ Halbrook was supported by F32CA228328, K99/R00CA241357, P30CA062203, and the Sky Foundation. The funders did not have a role in the planning, execution, or writing of this study.

## Additional information

### Competing interests

Costas A Lyssiotis: has received consulting fees from Astellas Pharmaceuticals, Odyssey Therapeutics, and T-Knife Therapeutics, and is an inventor on patents pertaining to Kras regulated metabolic pathways, redox control pathways in pancreatic cancer, and targeting the GOT1-pathway as a therapeutic approach (US Patent No: 2015126580-A1, 05/07/2015; US Patent No: 20190136238,

05/09/2019; International Patent No: WO2013177426-A2, 04/23/2015). The other authors declare that no competing interests exist.

## Funding

| Funder | Grant reference number | Author |
| --- | --- | --- |
| National Institutes of Health | T32-GM007315 | Rosa E Menjivar |
| National Cancer Institute | F31-CA257533 | Rosa E Menjivar |
| National Institutes of Health | T32-HD007505 | Rosa E Menjivar |
| University of Michigan | Rackham Merit Fellowship | Rosa E Menjivar |
| American Cancer Society | Scholar grant | Marina Pasca di Magliano |
| Pancreatic Cancer Action Network | | Marina Pasca di Magliano |
| National Institutes of Health | U01-CA224145 | Marina Pasca di Magliano |
| National Institutes of Health | R01-CA151588 | Marina Pasca di Magliano |
| National Cancer Institute | R01-CA198074 | Marina Pasca di Magliano |
| National Cancer Institute | U01CA274154 | Marina Pasca di Magliano |
| National Cancer Institute | R01CA268426 | Marina Pasca di Magliano |
| National Cancer Institute | R37-CA237421 | Costas A Lyssiotis |
| National Cancer Institute | R01-CA248160 | Costas A Lyssiotis |
| National Cancer Institute | R01-CA244931 | Costas A Lyssiotis |
| National Cancer Institute | R01-CA247516 | Howard C Crawford |
| University of Michigan | Postdoctoral Pioneer Program | Zeribe C Nwosu |
| University of Michigan | Training Program in Organogenesis | Wenting Du |
| National Cancer Institute | T32-CA009676 | Katelyn L Donahue Daniel Salas-Escabillas |
| National Cancer Institute | T32-AI007413 | Hanna S Hong |
| National Institute of Diabetes and Digestive and Kidney Diseases | T32-DK094775 | Hanna S Hong |
| National Cancer Institute | F31-CA247037 | Ashley Velez-Delgado |
| National Institute of General Medical Sciences | T32-GM008353 | Ashley Velez-Delgado |
| National Institutes of Health | T32-AI007413 | Padma Kadiyala |
| American College of Gastroenterology | T32-DK094775 | Eileen Carpenter |
| National Cancer Institute | R50-CA232985 | Yaqing Zhang |
| National Cancer Institute | F32-CA228328 | Christopher J Halbrook |
| National Institutes of Health | R00-CA241357 | Christopher J Halbrook |
| Sky Foundation | | Christopher J Halbrook |

The funders had no role in study design, data collection and interpretation, or the decision to submit the work for publication.

## Author contributions

Rosa E Menjivar, Conceptualization, Data curation, Formal analysis, Investigation, Methodology, Writing – original draft, Project administration; Zeribe C Nwosu, Formal analysis, Investigation, Writing – review and editing; Wenting Du, Hanna S Hong, Carlos Espinoza, Kristee Brown, Ashley Velez-Delgado, Wei Yan, Fatima Lima, Allison Bischoff, Padma Kadiyala, Daniel Salas-Escabillas, Christopher J Halbrook, Investigation; Katelyn L Donahue, Eileen Carpenter, Formal analysis; Howard C Crawford, Resources, Writing – review and editing; Filip Bednar, Supervision; Yaqing Zhang, Supervision, Investigation, Writing – review and editing; Costas A Lyssiotis, Resources, Supervision, Writing – review and editing; Marina Pasca di Magliano, Conceptualization, Resources, Supervision, Funding acquisition, Methodology, Project administration, Writing – review and editing

## Author ORCIDs

Rosa E Menjivar http://orcid.org/0000-0003-1551-869X
Allison Bischoff http://orcid.org/0000-0001-5119-5272
Daniel Salas-Escabillas http://orcid.org/0000-0002-0819-989X
Eileen Carpenter http://orcid.org/0000-0001-6775-6943
Costas A Lyssiotis http://orcid.org/0000-0001-9309-6141
Marina Pasca di Magliano http://orcid.org/0000-0001-9632-9035

## Ethics

Human research was performed in accordance with the Declaration of Helsinki and the ethical standards and guidelines approved by the University of Michigan Institutional Review Board. Patients provided written informed consent.

All the animal studies and procedures were conducted in compliance with the guidelines of the Institutional Animal Care and Use Committee (IACUC) at the University of Michigan, protocol number: PRO00009814.

## Decision letter and Author response

Decision letter https://doi.org/10.7554/eLife.80721.sa1
Author response https://doi.org/10.7554/eLife.80721.sa2

---

# Additional files

## Supplementary files
- MDAR checklist

## Data availability

Human sc-RNA-seq data was previously published (*Steele et al., 2020*) and both raw and processed data are available at the NIH dbGap database accession number phs002071.v1.p1. Raw and processed sc-RNA-seq data for the WT and KPC were previously published and are available at GEO accession number GSM5011580 and GSE202651. Raw and processed sc-RNA-seq data for the KF and KFCA are available at GEO accession number GSE203016.

The following dataset was generated:

| Author(s) | Year | Dataset title | Dataset URL | Database and Identifier |
|---|---|---|---|---|
| di Magliano MP, Menjivar RE, Donahue KL | 2022 | Arginase 1 deletion in myeloid cells decreases immune suppression and tumor formation in pancreatic cancer | https://www.ncbi.nlm.nih.gov/geo/query/acc.cgi?acc=GSE203016 | NCBI Gene Expression Omnibus, GSE203016 |

The following previously published datasets were used:

| Author(s) | Year | Dataset title | Dataset URL | Database and Identifier |
|---|---|---|---|---|
| di Magliano MP | 2020 | Multimodal Mapping of the Tumor and Peripheral Blood Immune Landscape in Human Pancreatic Cancer | https://www.ncbi.nlm.nih.gov/geo/query/acc.cgi?acc=GSE155698 | NCBI Gene Expression Omnibus, GSE155698 |
| di Magliano MP, Crawford HC, Kemp SB | 2021 | Pancreatic cancer is marked by complement-high blood monocytes and tumor-associated macrophages | https://www.ncbi.nlm.nih.gov/geo/query/acc.cgi?acc=GSM5011580 | NCBI Gene Expression Omnibus, GSM5011580 |
| di Magliano MP, Donahue KL, Steele NG | 2022 | Murine models of pancreatic cancer: KPC | https://www.ncbi.nlm.nih.gov/geo/query/acc.cgi?acc=GSE202651 | NCBI Gene Expression Omnibus, GSE202651 |

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
