## [Editor Report]

Menjivar et al. identify an important, previously unrecognized role of myeloid cell Arginase1 (Arg1) activity in shaping the anti-tumor immune response in pancreatic ductal adenocarcinoma (PDAC). The proposed therapeutic combination is a convincing new approach for pancreatic cancer, with an enhanced response to immune therapy upon arginase inhibition.

---

## [Decision Letter]

**Decision letter after peer review:**

Thank you for submitting your article "Arginase 1 is a key driver of immune suppression in pancreatic cancer" for consideration by *eLife*. Your article has been reviewed by 3 peer reviewers, including Gina M DeNicola as Reviewing Editor and Reviewer #1, and the evaluation has been overseen by Wafik El-Deiry as the Senior Editor.

Essential revisions:

1) Please tone down the language regarding Arg1 being previously used as merely a marker of macrophage identity. A general role for Arg1 and arginine metabolism by myeloid cells in immunosuppression has already been established by multiple studies, including those cited by the authors, in multiple tumor types. The authors should modify this text to avoid overstating their findings.

2) Please provide additional data or moderate other claims as indicated where they are not justified by the data presented. The claim that Arg1 deletion in macrophages delayed the formation of invasive disease is not completely justified by the data presented.

3) Please address reviewer points that require clarifications, and add quantifications where requested.

*Reviewer #1 (Recommendations for the authors):*

The authors should further support the claim that Arg1 deletion in macrophages delayed the formation of invasive disease, or moderate their language.

For the claim that Arg1 deletion in macrophages delayed the formation of invasive disease, ideally, more mice could be added to the cohort, but considering that it takes so long for the mice to develop tumors the authors could consider implanting the KPC 7940B line into LysMCre/+; Arg1fl/fl mice and performing a survival or time point analysis to further support this claim. Alternatively, they could perform this experiment with 7940B transplantation into BL6 mice and treat with the arginase inhibitor, then monitor the onset of PDAC.

*Reviewer #2 (Recommendations for the authors):*

Although this does not decrease the translational relevance of their findings, a main weakness of the manuscript is that the authors do not show what breaking down arginine and generating ornithine and urea does/is needed for metabolically to impact immunity, as no such metabolic assays or experiments are performed anywhere in their work.

1. It would therefore be best if the authors at least show that arginine breakdown is needed for CD8^+^ T cell activation and proliferation, as proposed/speculated.

A few additional comments:

2. CB-1158 inhibits both Arg1 and Arg2 as mentioned in the manuscript. Given that a metabolic role of tumor-cell specific Arg2 expression has previously been demonstrated in PDAC tumors, the observed anti-tumor CB-1158 effect could not be merely attributed to Arg1/Arg2 role in immunity, but also to tumor intrinsic effects of Arg2 on PDAC growth.

3. Figure 1A: ISH- co-IF staining data should be strengthened to support the claim; Zoomed in the photo shows 3 cells (yellow) that co-stain for ARG1(red) and CD45 (green). No quantification is provided. To support their claim "We observed prevalent ARG1 expression in CD45+ cells, and occasional low expression in ECAD+ cells", the authors should quantify the ISH/co-IF staining as done in other Figure 2 and its Supplement. It would be best if the authors could also show the individual fluorescence channels (prior to merging) in Supplementary data if they do not fit in the main figure.

4. The same comment as in point 3, i.e. quantify- for all other ISH-coIF figures, including Figure 1G; Figure Supplement 1E; Figure 2 Supplement 2H, and Figure 3G.

*Reviewer #3 (Recommendations for the authors):*

1. Mechanism of Arg1 upregulation in cancer cells: Inherent differences between spontaneous Kras-driven model and orthotopic Kras- p53-driven model should be discussed as a contextual backdrop for the potential role of Arg1. It is not clear from the data provided whether the determinant of Arg1 expression in cancer cells is driven by accumulating mutational burden (higher at baseline in aggressive KPC orthotopic model) vs time to progression as would be the case in many months that elapsed between PanIN and PDAC stages in KC model. Do orthotopic KPC lesions contain Tuft cells? Some quantification of data in Figure S4A may help as well.

2. Arginase 1 inhibitor activity in TME: are there changes in arginine levels in the TME of mice systemically treated with the inhibitor?

3. The authors should indicate in the main text the instances where the data shows the non-significant trend in PanIN frequency (ie. Figure S2C, page 16, line 316 currently described as 'decrease'); I would also strongly suggest making it clear that there are limitations with regards to sample size for scRNAseq analysis (one sample for each condition) and to take any conclusions that come with the description of said data with caution.

---

## [Author Response]

Essential revisions:1) Please tone down the language regarding Arg1 being previously used as merely a marker of macrophage identity. A general role for Arg1 and arginine metabolism by myeloid cells in immunosuppression has already been established by multiple studies, including those cited by the authors, in multiple tumor types. The authors should modify this text to avoid overstating their findings.

We have modified the text to better reflect our findings in the context of previous literature.

2) Please provide additional data or moderate other claims as indicated where they are not justified by the data presented. The claim that Arg1 deletion in macrophages delayed the formation of invasive disease is not completely justified by the data presented.

In our initial submission, we only had 12 mice for the KF group and 8 mice for the KFCA group, a limitation given the complexity of genotypes and the timeframe of the experiment (11-12 months). Several more mice were aging. In this resubmission, we include 26 and 24 mice, respectively; with the increase in animal numbers, our data indicating a delay in progression is substantiated. We have modified Figure 2E to reflect the new data as well as the related text on page 18.

3) Please address reviewer points that require clarifications, and add quantifications where requested.

We have amended the text for clarification and have included quantifications as requested by the reviewers. For Figures 1A and 1H, quantification is in Figure 1B and 1I, respectively. For Figure 2 Supplement 3D, quantification is in Figure 2 Supplement 3E. Quantification for Figure 3G is in Figure 3H.

Reviewer #1 (Recommendations for the authors):The authors should further support the claim that Arg1 deletion in macrophages delayed the formation of invasive disease, or moderate their language.For the claim that Arg1 deletion in macrophages delayed the formation of invasive disease, ideally, more mice could be added to the cohort, but considering that it takes so long for the mice to develop tumors the authors could consider implanting the KPC 7940B line into LysMCre/+; Arg1fl/fl mice and performing a survival or time point analysis to further support this claim. Alternatively, they could perform this experiment with 7940B transplantation into BL6 mice and treat with the arginase inhibitor, then monitor the onset of PDAC.

We thank reviewer 1 for these suggestions. Being aware that our initial numbers were small, we had been aging KF and KFCA mice to increase our cohorts. We now have 26 KF mice and 24 KFCA mice (compared with 12 and 8 previously). We have now included statistical analysis using Chi square test, showing statistical significance of the difference in tumor formation in the two cohorts (Figure 2E). Our results are consistent with the notion that deletion of Arginase 1 in myeloid cells decreases tumor formation in PDA.

Reviewer #2 (Recommendations for the authors):Although this does not decrease the translational relevance of their findings, a main weakness of the manuscript is that the authors do not show what breaking down arginine and generating ornithine and urea does/is needed for metabolically to impact immunity, as no such metabolic assays or experiments are performed anywhere in their work.1. It would therefore be best if the authors at least show that arginine breakdown is needed for CD8^+^ T cell activation and proliferation, as proposed/speculated.

Reviewer 2 raised a very important point, namely how, mechanistically, loss of Arginase increases T cell numbers in the tumor. Arginase breaks down arginine into ornithine and urea. Thus, it is equally conceivable that lack of arginine (Arg), or excess of ornithine (Orn) and/or urea, might impair the ability of CD8+ T cells to proliferate and/or become activated. In fact, a recent preprint from the Weaver group at UCSF (doi.org/10.1101/2022.07.14.499764) suggests that, in the context of breast cancer, Ornithine might directly impair T cell activation.

We designed the following experiment: we isolated Naive CD8+ T cells from spleens and lymph nodes of C57BL/6 mice and cultured them for 72 hrs with complete RPMI media or with media lacking arginine (-Arg RPMI). In both media conditions, we added groups where the media was supplemented with ornithine (Orn), with urea, or with a combination of ornithine and urea. We show that arginine deprivation results in reduced CD8 T cell proliferation and activation. We found that culturing cells in RPMI media containing urea slightly enhanced T cell proliferation and activation compared to RPMI media control, while no enhancement was observed with ornithine supplementation. When cells were cultured in RPMI media lacking arginine, supplementation with ornithine, but not urea, partially rescued the proliferation defect caused by lack of arginine. No further rescue occurred adding Urea; neither ornithine nor urea rescued the activation defect caused by lack of arginine. We thus concluded that CD8 T cell require arginine for proliferation and activation, and that the arginine breakdown byproducts are not harmful to CD8 T cell proliferation nor are they able to fully rescue lack of arginine. The new data is included the revised manuscript in Figure 4I, J, and K. Text describing these findings is included on page 32 and 33 of the revised manuscript.

A few additional comments:2. CB-1158 inhibits both Arg1 and Arg2 as mentioned in the manuscript. Given that a metabolic role of tumor-cell specific Arg2 expression has previously been demonstrated in PDAC tumors, the observed anti-tumor CB-1158 effect could not be merely attributed to Arg1/Arg2 role in immunity, but also to tumor intrinsic effects of Arg2 on PDAC growth.

This is an important point. Arg2 is upregulated in obesity-driven PDA and silencing, or loss of Arg2 in PDA tumor cells suppressed tumor growth in obesity-associated PDA (Zaytouni et al., 2017). In our study, we found that the most highly expressed *Arginase* gene in both mouse and human PDA was Arg1 and not Arg2 (Figure 1). However, upon deletion of *Arg1* in myeloid cells, we observed upregulation of *Arg1* in epithelial cells and upregulation of *Arg2* in a subset of macrophages, both likely compensatory mechanisms (Figure 2F and Figure 2—figure supplement 2A, and Figure 2I). Thus, it is also possible that the suppression in tumor growth upon treatment with the Arginase inhibitor is due to the systemic inhibition of both Arginase isoforms in multiple cell compartments, including epithelial cells. We have amended the text in the Discussion section on page 43 to acknowledge this possibility.

3. Figure 1A: ISH- co-IF staining data should be strengthened to support the claim; Zoomed in the photo shows 3 cells (yellow) that co-stain for ARG1(red) and CD45 (green). No quantification is provided. To support their claim "We observed prevalent ARG1 expression in CD45+ cells, and occasional low expression in ECAD+ cells", the authors should quantify the ISH/co-IF staining as done in other Figure 2 and its Supplement. It would be best if the authors could also show the individual fluorescence channels (prior to merging) in Supplementary data if they do not fit in the main figure.

We have now included quantification of the *ARG1* RNA scope (in situ hybridization) in combination with immunofluorescence staining for CD45 and ECAD. We have also included the individual fluorescence channels as recommended by reviewer 2. This data supports the claim that there is prevalent *ARG1* expression in CD45^+^ cells and occasional expression in ECAD^+^ cells. See Figure 1A and 1B.

4. The same comment as in point 3, i.e. quantify- for all other ISH-coIF figures, including Figure 1G; Figure Supplement 1E; Figure 2 Supplement 2H, and Figure 3G.

We have now added quantification to the immunofluorescent stainings. Figure 1G is now 1H. The quantification for Figure 1—figure supplement 1E is included in Figure 1I. Statistical significance was determined using a one-way ANOVA with multiple comparisons. The quantification for Figure 2 Supplement 2H, now Figure 2 Supplement 3D is shown in Figure 2 Supplement 2E, and the quantification for Figure 3G is shown in Figure 3H. Student’s t test was used to determine significance.

Reviewer #3 (Recommendations for the authors):1. Mechanism of Arg1 upregulation in cancer cells: Inherent differences between spontaneous Kras-driven model and orthotopic Kras- p53-driven model should be discussed as a contextual backdrop for the potential role of Arg1. It is not clear from the data provided whether the determinant of Arg1 expression in cancer cells is driven by accumulating mutational burden (higher at baseline in aggressive KPC orthotopic model) vs time to progression as would be the case in many months that elapsed between PanIN and PDAC stages in KC model. Do orthotopic KPC lesions contain Tuft cells? Some quantification of data in Figure S4A may help as well.

Tuft cells emerge during early lesions as a response to injury, but progressively decrease in advanced PanINs and invasive PDA (Bailey et al., 2014; Delgiorno et al., 2014); thus, they are not found in orthotopic tumors; we clarified this in the text.

2. Arginase 1 inhibitor activity in TME: are there changes in arginine levels in the TME of mice systemically treated with the inhibitor?

We performed metabolomic analysis in the tumors treated with vehicle, arginase inhibitor, anti-PD1, and with the combination of arginase inhibitor and antiPD1 and observed a significant increase in arginine levels in the combination group. Data is shown in manuscript Figure 4H.

3. The authors should indicate in the main text the instances where the data shows the non-significant trend in PanIN frequency (ie. Figure S2C, page 16, line 316 currently described as 'decrease'); I would also strongly suggest making it clear that there are limitations with regards to sample size for scRNAseq analysis (one sample for each condition) and to take any conclusions that come with the description of said data with caution.

We have amended the text on page 16, (page 17 and page 18 in the revised manuscript) in agreement with the reviewer’s suggestions.